# Seeing Beyond Classes: Zero-Shot Grounded Situation Recognition via Language Explainer

## ABSTRACT

Benefiting from strong generalization ability, pre-trained vision-language models (VLMs), *e.g.*, CLIP, have been widely utilized in zero-shot scene understanding. Unlike simple recognition tasks, grounded situation recognition (GSR) requires the model not only to classify salient activity (verb) in the image, but also to detect all semantic roles that participate in the action. This complex task usually involves three steps: verb recognition, semantic role grounding, and noun recognition. Directly employing class-based prompts with VLMs and grounding models for this task suffers from several limitations, *e.g.*, it struggles to distinguish ambiguous verb concepts, accurately localize roles with fixed verb-centric template[1] input, and achieve context-aware noun predictions. In this paper, we argue that these limitations stem from the model's poor understanding of verb/noun *classes*. To this end, we introduce a new approach for zero-shot GSR via Language EXplainer (LEX), which significantly boosts the model's comprehensive capabilities through three explainers: 1) verb explainer, which generates general verb-centric descriptions to enhance the discriminability of different verb classes; 2) grounding explainer, which rephrases verb-centric templates for clearer understanding, thereby enhancing precise semantic role localization; and 3) noun explainer, which creates scene-specific noun descriptions to ensure context-aware noun recognition. By equipping each step of the GSR process with an auxiliary explainer, LEX facilitates complex scene understanding in real-world scenarios. Our extensive validations on the SWiG dataset demonstrate LEX's effectiveness and interoperability in zero-shot GSR.

## CCS CONCEPTS

• **Computing methodologies** → **Scene understanding**.

## KEYWORDS

Zero-Shot GSR, Vision-Language Model, Large Language Model

## 1 INTRODUCTION

Traditional recognition methods heavily rely on the quality of vast dataset annotations, often limited in generalizing across diverse or unseen scenarios [35]. Zero-shot learning emerges to enable models to identify classes they have never been directly trained

---

[1]The class-contained/verb-centric template contains both verb class and its associated semantic roles, which is provided in the dataset.

*ACM MM, 2024, Melbourne, Australia*

© 2024 Copyright held by the owner/author(s). Publication rights licensed to ACM.
ACM ISBN 978-x-xxxx-xxxx-x/YY/MM
https://doi.org/10.1145/nnnnnnn.nnnnnnn

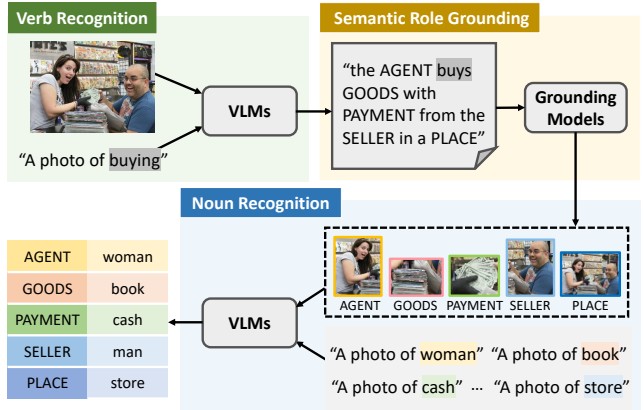

**Figure 1: Illustration of the straightforward pipeline for Zero-Shot GSR. 1) Verb Recognition: utilizing VLMs to identify verbs via verb class-based prompts. 2) Semantic Role Grounding: employing grounding models to localize semantic roles based on the verb-centric template[1] . 3) Noun Recognition: applying VLMs for identifying entities within localized semantic roles through noun class-based comparison.**

on, thus greatly enhancing their universality in real-world settings [38, 43, 44]. Recently, advancements in pre-trained vision-language models (VLMs) [19, 20, 27], *e.g.*, CLIP [37], have shown excellent generalization capabilities across various recognition tasks, setting new benchmarks in zero-shot learning. Particularly, CLIP incorporates dual encoders: an image encoder and a text encoder. The former processes visual input into visual features, and the latter translates texts into semantic features. This architecture facilitates the alignment of visual and text data within a unified semantic space. Leveraging class-based prompts like "A photo of [NOUN CLASS]" or "A photo of [VERB CLASS]", CLIP effectively compares images against prompts in the learned semantic space, enabling the zero-shot recognition of both unseen objects [37] and actions [29].

Nevertheless, training-free zero-shot grounded situation recognition (ZS-GSR) presents a more intricate challenge than basic object or action recognition [36]. It requires not only identifying actions (verbs) depicted in images but also discerning and localizing the semantic roles involved [36, 42]. Such a comprehensive task contributes to effectively understanding **what** is happening (*e.g.*, *buying* in Figure 1), **who** is involved (*e.g.*, *AGENT* and *SELLER*), **where** it is taking place (*e.g.*, *PLACE*), *etc.* Consequently, ZS-GSR extends beyond simple **class** recognition to a structured understanding of scenes, requiring accurate modeling of the relationships among objects within an event. As illustrated in Figure 1, a typical GSR approach typically consists of three steps:

- **Verb Recognition.** This step aims to identify the verb category of the entire image. The most common zero-shot method for this sub-task uses a set of class-based prompts "a photo of [VERB

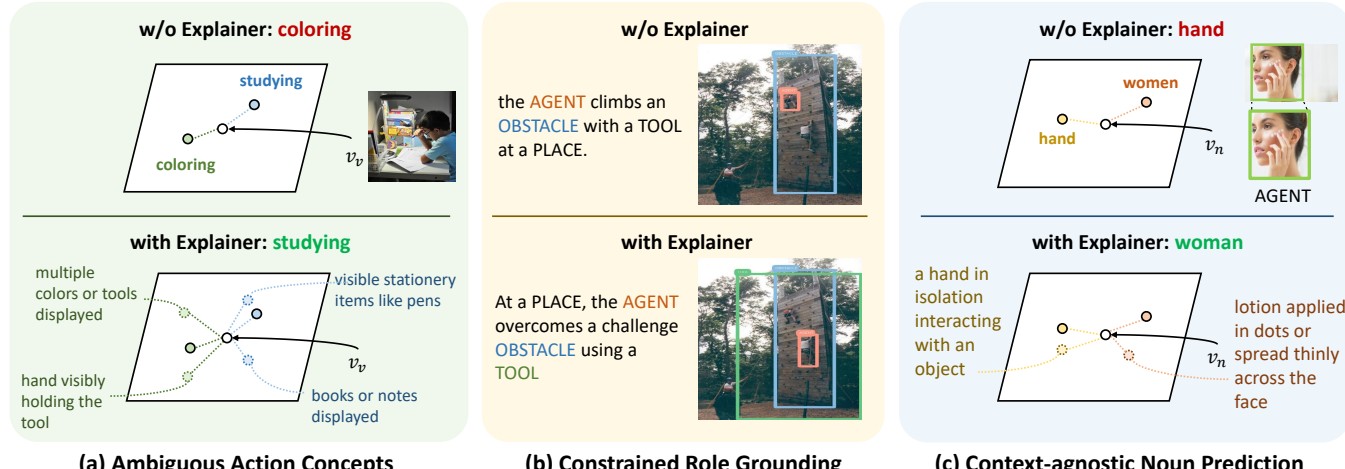

**Figure 2: The limitations of class-based prompts for zero-shot GSR. (a) Ambiguous Action Concepts: the verb "studying" is mistakenly identified as "coloring" due to an unclear verb meaning. (b) Constrained Role Grounding: Rigid templates misguide the grounding of "TOOL" in a complex scene. (c) Context-agnostic Noun Prediction: the noun "woman" is incorrectly classified as "hand" without considering semantic role context.**

CLASS]" for each verb category [37]. These prompts are then fed into the text encoder to obtain semantic embedding, while the whole image is passed through an image encoder to get visual embedding. By comparing these two embeddings in the same semantic space, CLIP can achieve zero-shot verb recognition.

- **Semantic Role Grounding.** Given the predicted verb class and its predefined template[1], this step aims to precisely localize each semantic role within the image. Existing methods for this sub-task often adopt an open-world grounding model, *e.g.*, Grounding DINO [30], with an object-contained grounding language as input [17, 49, 54]. As for GSR, the input can be the provided templates in the dataset, *e.g.*, the template of *buys*: "the AGENT buys GOODS with PAYMENT from the SELLER in a PLACE".

- **Noun Recognition.** This step aims to identify the noun category for each semantic role. Similar to the verb, a most direct zero-shot method can use noun class-based prompts like "A photo of {NOUN CLASS}" to compare with visual features of grounded semantic role for classification [37]. For example in Figure 1, the grounded semantic role *AGENT* is classified as *woman*.

However, these simple methods that rely entirely on class-based prompts at every step are limited by: 1) **Ambiguous Action Concepts**: Foundational VLMs primarily concentrate on understanding images or bag of entities, often overlooking the semantics and structures inherent in actions [10, 26, 28]. Directly classifying verbs with class-based prompts may not fully capture the nuanced meanings of actions, leading to CLIP misunderstanding the verb's embedding and generating inaccurate predictions. For instance in Figure 2 (a), the model projects the two embeddings into adjacent locations in semantic space and recognizes an activity as "*coloring*" due to the colorful background. It overlooks the actual activity of "*studying*" that entails a complex interaction with stationery and books. 2) **Constrained Role Grounding**: The effectiveness of semantic role grounding within visual grounding models critically depends on the quality of the text prompt [39, 46]. The employment of fixed and

simplex templates[1] for semantic role grounding is inherently constrained by the simplified sentence structure and ambiguous verb. Once the model encounters unfamiliar or complex verb classes, it easily leads to misalignment and localization errors for the involved roles. As depicted in Figure 2 (b), with templates centered on the unfamiliar "*climb*" class, grounding models have difficulty in locating the "*TOOL*". 3) **Context-agnostic Noun Prediction**: Noun prediction is often conditioned on the specific verb and semantic role present within a scene [36, 42]. However, when employing class-based prompts for noun classification, CLIP tends to concentrate on the category with the highest prediction confidence. This method overlooks the crucial consideration of whether the noun accurately fits the specific role defined in the scene's template, leading to contextually inappropriate predictions. For example, in Figure 2 (c), VLMs directly recognize the "*hand*" that occupies the main area of the image, ignoring the characteristics of the *AGENT*.

In this paper, we argue that these limitations stem from the model's insufficient understanding of **individual classes**. When humans encounter an unfamiliar word, such as "soliloquy", we often turn to a dictionary to serve as an explainer, providing a clear and accessible definition that illuminates the term's meaning from a common understanding perspective. For instance, the term "soliloquy" in a theatrical script refers to "a character speaking their thoughts aloud when alone or regardless of any hearers". Fortunately, recent advancements in large language models (LLMs), *e.g.*, GPT [4], have benefited from extensive training across diverse datasets, endowing them with broad world knowledge.

Inspired by human reliance on external sources for deeper understanding, we propose the method for zero-shot grounded situation recognition via **L**anguage **EX**plainer (**LEX**). It employs LLMs to serve as explainers at crucial steps of the GSR process. Specially, for verb recognition, we devise a **verb explainer** ♙ to prompt LLMs to generate general verb-centric descriptions. This approach provides CLIP with multiple enriched "explanations" of each verb, *e.g.*,

"books or notes displayed" in Figure 2 (a), leading to more accurate verb classification. Additionally, we implement an offline description weighting strategy that takes into account the discriminability of the verb-centric descriptions without any training data. As for semantic role localization, we design a **grounding explainer** 🏺 to prompt the LLMs to rephrase the original verb-centric template from multiple perspectives. These reformulated sentences, as seen in Figure 2 (b), serving as text inputs for the grounding model, are capable of enhancing the model's understanding of the intricate relationships between semantic roles. Lastly, for noun recognition, we propose a **noun explainer** 🏺 to prompt LLMs to generate noun descriptions conditioned on both verb and semantic roles. As is illustrated in Figure 2 (c), noun explainer prevents the misclassification of a noun such as "*hand*" when the contextual role and action pertain to "*woman*" applying lotion, thus mitigating the risk of contextually inappropriate classifications.

To verify the effectiveness of our LEX, we conduct extensive experiments and ablation studies on SWiG dataset. Each of our explainers can be utilized as a plug-and-play module to improve the overall performance of zero-shot GSR.

In summary, we make the following contributions in this paper:

(1) We propose a novel training-free zero-shot GSR framework LEX that incorporates LLMs as explainers within each process, enhancing understanding of complex visual scenes.
(2) We introduce three explainers: verb explainer, grounding explainer, and noun explainer. Each serves as a plug-and-play module to enhance the performance of zero-shot GSR.
(3) We devise a strategy to weight descriptions by discriminability, independent of training data.
(4) Extensive experiments on the SWiG dataset demonstrate the effectiveness and interpretability of LEX.

## 2 RELATED WORK

**Grounded Situation Recognition (GSR)**. Despite deep learning's notable achievements in image classification [7, 31, 34], object detection [55], and image segmentation [6, 22], its comprehension of complex scenes remains limited. Techniques, *e.g.*, scene graph generation [25, 45, 52] and human-object interaction detection [1, 5, 16], aim to parse scene contents via relation graphs. Admittedly, these tasks can provide structured visual scene representations to assist downstream tasks [23]. However, they only focus on modeling with dyadic relationships between a subject and an object, ignoring the diversity of roles in visual events [8]. GSR emerges as a solution for holistic scene understanding by predicting actions (verbs) and detecting noun entities of each semantic role [36]. Pratt *et.al.* [36] first proposed an RNN-based two-stage framework, which detects verbs in the first stage and predicts the nouns with bounding boxes in the second stage. Subsequent methods[8, 10, 42] improved this two-stage pipeline by employing a transformer-based model and considering semantic relations among verbs and semantic roles. However, previous efforts that depend on annotated training samples may face challenges like sample noises [23, 24] and long-tailed distribution [9, 40, 48], potentially limiting real-world applicability. Our approach predicts verbs, nouns, and their associated bounding boxes without training data, showcasing robust generalization and interpretability in real-world scenarios.

**Foundation Models**. Foundation models are typically pre-trained on extensive training data. Owing to their strong generalization capabilities, foundation models are often applied to a variety of downstream tasks [3]. Particularly in the natural language processing (NLP) area, large language models (LLMs) [11, 12, 18], such as GPT-3 [14], PaLM [2], OPT [53], and LLaMA [41], trained on extensive web-scale text datasets, exhibit formidable capabilities ranging from text prediction to contextually relevant text generation. In the field of cross-modal research, vision-language models (VLMs), *e.g.*, CLIP [37] and ALBEF [21], have been trained on extensive image-text pairs via contrastive learning, facilitating the cohesive integration of information between these modalities. Subsequent models, such as BLIP [20] and BLIP-2 [19], leverage diverse datasets and Q-Former's trainable query vectors to further refine this alignment. Despite the remarkable transferability of VLMs, prevailing approaches often rely on class-based hard prompts as input to the text encoder, leading to a narrow focus on specific visual features and disregarding contextual information in the surroundings. In this paper, we employ various prompts to guide LLMs as different explainers to generating "class explanations", significantly enhancing VLM's understanding of intricate scenes.

## 3 ZS-GSR VIA LANGUAGE EXPLAINER

**Formulation.** Given an image $I$, zero-shot GSR aims to identify a structured frame $\mathcal{F}_v = \{y_v, \mathcal{FR}_v\}$, where $y_v \in \mathcal{V}$ denotes the salient verb category, $\mathcal{FR}_v = \{(r, y_n, \mathbf{b}) \mid r \in \mathcal{R}_v\}$ represents the set of filled semantic roles. The $\mathcal{R}_v$ represents the set of predefined semantic roles associated with verb-centric template $TP_v$. Each role $r \in \mathcal{R}_v$ is filled with a noun $y_n \in \mathcal{N}$ grounded by a bounding box $\mathbf{b} \in \mathcal{B}$. For instance in Figure 1, the frame can be detected as $\mathcal{F}_v =$ {*buying*, {(*AGENT*, *woman*, □), (*GOODS*, *book*, □), (*PAYMENT*, *cash*, □), (*SELLER*, *man*, □), (*PLACE*, *store*, □)}}.

**Baseline for Zero-Shot GSR.** As mentioned above, the baseline involves three steps: *1) Verb Recognition*, the class-based prompts is fed into text encoder $T(\cdot)$ of CLIP to obtain verb text features $\{\mathbf{t}_v\}$. Similarly, the entire image is fed into image encoder $V(\cdot)$ to extract visual feature $\mathbf{v}_v$. Subsequently, the cosine similarity between $\mathbf{t}_v$ and $\mathbf{v}_v$ is computed across different verb categories for final prediction. *2) Semantic Role Grounding*, Grounding DINO integrates a verb-centric template $TP_v$ as text input and image $I$ as visual input to generate a series of candidate bounding boxes with semantic role labels and scores. The bounding boxes with the highest score for each semantic role are selected. *3) Noun Recognition*, CLIP compares cropped region feature $\{\mathbf{v}_n^r\}$ of each role $r$ with the text features $\{\mathbf{t}_n\}$ of class-based prompts for noun classification.

To address the constraints of the baseline, we propose a new framework LEX for ZS-GSR. As illustrated in Figure 3, it comprises three components: verb recognition via verb explainer, role localization via grounding explainer, and noun recognition via noun explainer. LEX enhances the model's understanding of different classes by adding an auxiliary explainer at each step.

### 3.1 Verb Recognition via Verb Explainer

This component aims to recognize the salient verb category within an image. It comprises three steps: verb-centric description generation, description weighting, and verb classification.

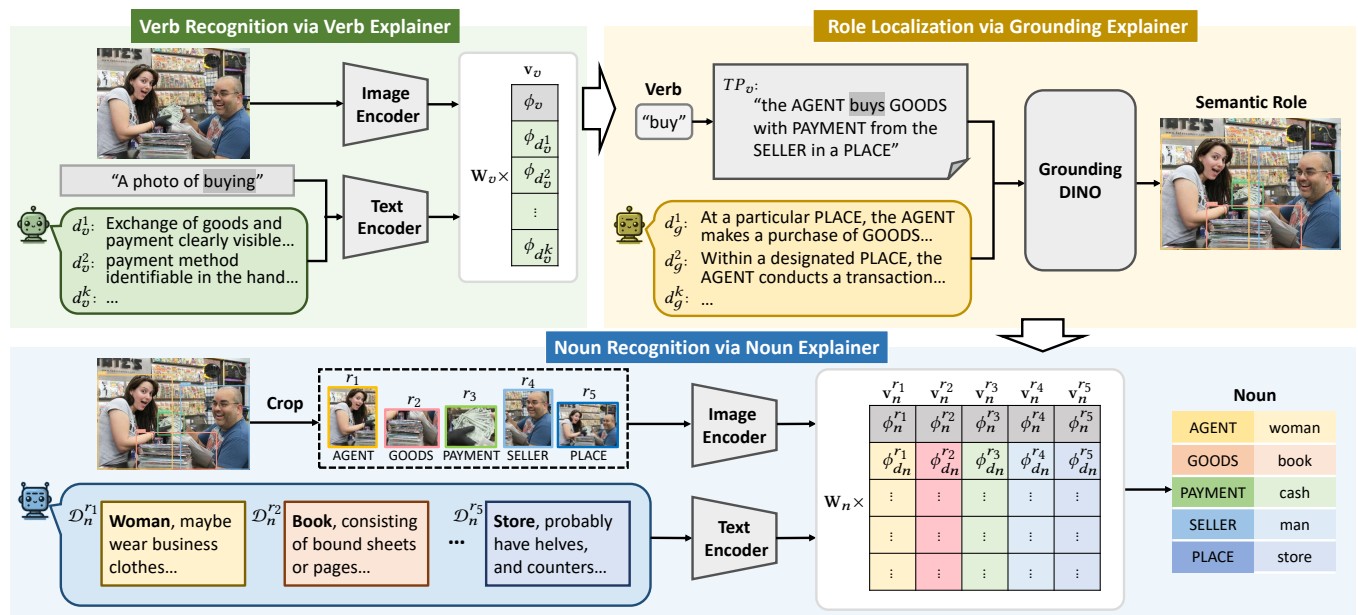

**Figure 3: The framework of LEX. 1) Verb Recognition via Verb Explainer: generate general verb-centric descriptions to recognize verbs. 2) Role Localization via Grounding Explainer: generate a rephrased verb-centric template to localize semantic roles. 3) Noun Recognition via Noun Explainer: generate scene-specific noun descriptions to predict nouns.**

*3.1.1 Verb-Centric Description Generation.* To enhance the discriminative capacity of the CLIP model in distinguishing verb categories, we introduce a verb explainer to "explain" basic class-based prompts for each category. Inspired by zero-shot image classification [32], we introduce a verb-centric description generation prompt to make LLMs as verb explainer to generate general verb-centric descriptions $\mathcal{D}_v = \{d_v\}$ for verb class $y_v \in \mathcal{V}$ from multiple perspectives:

$$\mathcal{D}_v = LLM\ \underbrace{(\text{in-context examples}, y_v, \text{instruction})}_{\text{prompt input}}, \quad (1)$$

where $LLM(\cdot)$ is the decoder of the LLM. The verb-centric description generation prompt input consists of three parts[2]: 1) *In-context examples*, the description instruction for the verb, along with some examples of the generated results. This part adopts in-context learning [13, 33] to make LLM generate analogous results. 2) *Verb class* $y_v$, the target category that needs to generate the description. 3) *Instruction*, the sentence used to command the LLM to generate description, *e.g.*, "what are the useful visual features for the event of 'rehabilitating': AGENT rehabilitates ITEM at a PLACE".

These generated descriptions highlight various unique visual features associated with specific action categories, for example, "looking around for searching goods", "pushing a shopping cart or carrying a basket", contribute to enhancing the distinguishability between "*shopping*" and other similar actions like "*buying*".

*3.1.2 Description Weighting.* Considering the variances in description quality and its different impact on verb classification, we devise a description weighting strategy. Recent methods try to select descriptions from the perspective of the discrimination of image

---

[2]The detailed description generation prompts are in the Appendix.

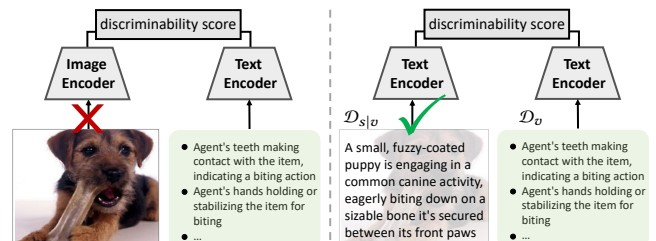

**Figure 4: An example of using "biting"'s scene text as "image" to calculate the discriminability score.**

features [47] and the coverage among concepts [50]. However, these methods require a large number of annotated training images and are prone to overfitting to seen scenarios, which are not suitable for training-free zero-shot GSR. Thanks to the property of VLMs to align visual and text modes in a shared space, we can ***replace an annotated image with a complex scene description that contains the verb category*** [15].

Specifically, we first utilize the LLMs to generate a set of complex scene descriptions $\mathcal{D}_{s|v} = \{d_s\}$ for each verb $y_v \in \mathcal{V}$. These scene descriptions are fed into the text encoder $T(\cdot)$ to generate scene text embeddings $\{\mathbf{t}_s\}$, used as "image" for subsequent description weighting[2]. For instance in Figure 4, the verb *biting*'s scene description is used instead of an annotated image.

Intuitively, the greater the distinctiveness of a verb description in differentiating between its own class of scenes and other classes, the more important it is. Following [47], we apply the **discriminability score** $Dis(\cdot)$ to measure the distinctiveness of a verb description. The correlation $\rho(\cdot, \cdot)$ between a verb description embedding and a

set of scene text embeddings with class $y_v$ is denoted as:

$$\rho(y_v, d_v) = \frac{1}{|\mathcal{D}_{s|v}|} \sum_{d_s \in \mathcal{D}_{s|v}} \phi(\mathbf{t}_s, \mathbf{t}_{d_v}), \tag{2}$$

where $\phi(\cdot, \cdot)$ denotes the cosine similarity, $\mathbf{t}_{d_v}$ is the text embedding of a verb description $d_v$. The conditional likelihood of aligning scenes of a class given $d_v$ can be written as:

$$\overline{\rho}(y_v, d_v) = \frac{\rho(y_v, d_v)}{\sum_{y'_v \in \mathcal{V}} \rho(y'_v, d_v)}. \tag{3}$$

The discriminability score $Dis(d_v)$ can be measured by the entropy over all verb classes, written as:

$$Dis(d_v) = - \sum_{y'_v \in \mathcal{V}} \overline{\rho}(y'_v | d_v) \log(\overline{\rho}(y'_v | d_v)). \tag{4}$$

The smaller the discriminability score, the greater the distinction $d_v$ offers for its category. Hence, the weight of a general verb description can be denoted as:

$$w_v(d_v) = \frac{\exp\left(1/Dis(d_v)\right)}{\sum_{d'_v \in \mathcal{D}_v} \exp\left(1/Dis(d'_v)\right)}, \tag{5}$$

where $w_v(d_v)$ represents the weight of verb description $d_v$.

*3.1.3 Verb Classification.* In this step, we compute the similarity score between visual and textual features to obtain the probability distribution of verbs. We calculate the final distribution as follows:

$$Score(y_v) = \underbrace{(1 - \lambda)\phi(\mathbf{v}_v, \mathbf{t}_v)}_{\text{class-based}} + \underbrace{\lambda \sum_{d_v \in \mathcal{D}_v} w_v(d_v)\phi(\mathbf{v}_v, \mathbf{t}_{d_v})}_{\text{description-based}}, \tag{6}$$

where $\lambda$ is utilized as a balance factor between classed-based prompts and description-based prompts.

## 3.2 Role Localization via Grounding Explainer

This module aims to locate each semantic role more accurately via an auxiliary grounding explainer. Specifically, we first employ a grounding explainer to rephrase the verb-centric templates in a more comprehensible manner. Subsequently, we use the Grounding DINO to generate candidate bounding boxes for the semantic roles.

*3.2.1 Rephrased Template Generation.* Due to the ambiguity of the original fixed verb-centric template, using it as grounding text input for the Grounding DINO model may result in limited or inaccurate semantic role localizations. To enhance the comprehensibility of grounding language, we introduce a grounding explainer to "explain" these fixed templates. To be specific, we employ a rephrased template generation prompt, utilizing LLM as grounding explainers to produce rephrased grounding templates $\mathcal{D}_g = \{d_g\}$ for each verb class $y_v \in \mathcal{V}$, expressed as:

$$\mathcal{D}_g = LLM \underbrace{(\text{in-context examples}, TP_v, \text{instruction})}_{\text{prompt input}}. \tag{7}$$

Similarly, the rephrased template generation prompt input also comprises three components: in-context examples, template $TP_v$, and instruction[2]. The first two components are analogous to verb-centric description generation. As for the last component, the instruction

sentences are designed to enable LLM to generate role-contained templates that are easy to understand, *i.e.*, "Please generate new sentences that detailedly rephrase this sentence to make it easier to understand {TEMPLATE}. Note that you must keep the original words: {SEMANTIC ROLES}".

*3.2.2 Candidate Box Generation.* Given an unlabeled input image $I$ and the grounding text input $d_g$, the Grounding DINO model is adept at associating visual features with corresponding object labels (*i.e.*, semantic roles). More concretely, the Grounding DINO model outputs a sequence of bounding boxes $\mathcal{B}_g = \{\mathbf{b}\}$, associated object labels $\mathcal{R}_g = \{y_r\}$, and confidence scores $C_g = \{c\}$ as follows:

$$\mathcal{B}_g, \mathcal{R}_g, C_g = DINO(I, d_g), \tag{8}$$

where $DINO(\cdot)$ is the decoder of Grounding DINO. It is possible to generate multiple boxes for each object. We select the box with the highest confidence score $c$ for each object $y_r$ that appeared in the predefined role set $R_v$ as candidates. The chosen $\widehat{\mathcal{B}}_g$ of each grounding description $d_g$ constitutes the candidate boxes $\widehat{\mathcal{B}} = \{\widehat{\mathcal{B}}_g | d_g \in \mathcal{D}_g\}$. These candidate bounding boxes of semantic roles are utilized for the next noun recognition.

## 3.3 Noun Recognition via Noun Explainer

In this module, we utilize the noun explainer to achieve context-aware noun classification. To be specific, it consists of four steps: noun filtering, scene-specific noun description generation, noun pre-prediction, and noun refinement, as displayed in Figure 5.

*3.3.1 Noun Filtering.* This step filters out noun categories that are unlikely to appear in the specific semantic roles from class-level. Leveraging the common sense contained in LLM, we can inquire about reasonable noun categories through a simple prompt that includes a specified semantic role[2]. For instance, we can query "Given entity list {Noun CLASS}, which entities are most likely to be the result of a predicted semantic role PLACE" to filter unreasonable noun categories, *e.g.*, *man*. This filtering step not only enhances the computational efficiency of subsequent noun classification by reducing the number of traversed categories but also improves the accuracy of classification results.

*3.3.2 Scene-Specific Noun Description Generation.* In particular, the cropped candidate box inevitably contains more than one object, and the object normally has more than one possible category. Reliance on class-based prompts often results in CLIP focusing on the most salient noun category directly, thereby ignoring the specific scene context. To address this, we incorporate a noun explainer to "explain" class-based prompts associated with each noun category within diverse scenes[2]. This noun explainer adopts a scene-specific noun description generation prompt, guiding LLM to produce descriptions $\mathcal{D}^r_n = \{d_n\}$ that reflect the intricate context of each scene for each noun class $y_n \in \mathcal{N}$:

$$\mathcal{D}^r_n = LLM \underbrace{(\text{in-context examples}, \text{scene}, y_n, \text{instruction})}_{\text{prompt input}}. \tag{9}$$

Different from verb-centric description generation, the input prompt for scene-specific noun description generation is conditioned on extra scene information (verb-centric template and semantic role) to provide contextual information. The instruction is like this:

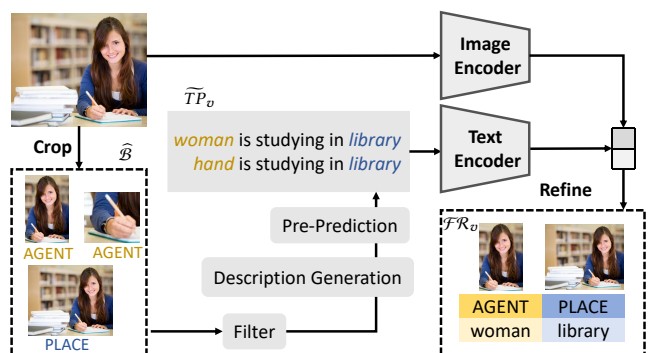

**Figure 5: The architecture of noun recognition.**

"Please describe the visual features that can distinguish the noun entity {NOUN CLASS} corresponding to {SEMANTIC ROLE} in the scene: AGENT writes on TARGET using a TOOL at a PLACE". The generated descriptions (*e.g.*, "holding a pen and other tools in hand") reflect the characteristic of corresponding semantic roles (*e.g.*, *AGENT*), thereby ensuring more accurate and contextually relevant noun categorization.

*3.3.3 Noun Pre-prediction.* In this step, we first compute the similarity between visual embeddings of all cropped boxes and text embedding to obtain the probability distribution of nouns. Similar to the verb classification process, this entails the incorporation of both class-based and description-based approaches to yield the pre-predict results for nouns:

$$Score(y_n) = \underbrace{(1-\lambda)\phi(\mathbf{v}_n^r, \mathbf{t}_n)}_{\text{class-based}} + \underbrace{\lambda \sum_{d_n \in \mathcal{D}_n^r} w_n(d_n)\phi(\mathbf{v}_n^r, \mathbf{t}_{d_n})}_{\text{description-based}}, \quad (10)$$

where $\mathbf{t}_{d_n}$ is the text features of the noun description $d_n$. Due to the requirement of invoking a large number[3] of LLMs for generating scene descriptions akin to description weighting in verb recognition, the weights $w_n(d_n)$ are uniformly assigned as $1/|\mathcal{D}_n^r|$.

*3.3.4 Noun Refinement.* After localization by Grounding DINO, a semantic role may correspond to multiple candidate boxes due to the integration of different $d_g \in \mathcal{D}_g$. Directly selecting the box with the highest confidence often relies solely on the localization model's understanding of the semantic role, overlooking the contextual relevance within the scene. To mitigate this, we propose to refine noun classification and bounding box selection via a global comparison. This process contrasts the visual features of the entire image $\mathbf{v}_v$ against the text features $\{\mathbf{t}_{tp}\}$ of templates $\{\widetilde{TP}_v\}$ whose semantic roles are filled iteratively with predicted nouns $\{y_n\}$ in the previous step, *e.g.*, "**woman** is studying in **library**" in Figure 5. The noun filled in the template with the highest similarity along with its corresponding bounding box is the final prediction, written as:

$$\widetilde{TP}_v^* = \arg\max \ \phi(\mathbf{v}_v, \{\mathbf{t}_{tp}\})$$
$$\mathcal{FR}_v = \{(r, y_n, b) \ | \ y_n \in \widetilde{TP}_v^*\}. \quad (11)$$

---

[3]Since noun description is conditioned on both verb and semantic role, the call number of LLMs at least is $|\mathcal{V}| \times |\mathcal{R}_v| \times |\mathcal{N}|$

As seen in Figure 5, the noun "*woman*" for the semantic role "*AGENT*" and its bounding box is selected as the final prediction result.

## 4 EXPERIMENT

### 4.1 Experimental Setup

*4.1.1 Dataset.* We evaluated our method on the SWiG dataset [36], which extends the imSitu dataset [51] by incorporating additional bounding box annotations for all visible semantic roles (69.3% of semantic roles have bounding boxes). In the SWiG dataset, each image is annotated with a verb, followed by a set of semantic roles ranging from 1 to 6 (3.55 on average), each verb is annotated with three verb frames by three separate annotators. SWiG contains 25200 testing images with 504 verb categories, 190 semantic role categories, and 9929 noun entity categories.

*4.1.2 Evaluation Metrics.* We employed the same five evaluation metrics as [36], which include: 1) **verb**: the accuracy of verb prediction. 2) **value**: the noun prediction accuracy for each semantic role. 3) **value-all (val-all)**: the noun prediction accuracy across the entire set of semantic roles. 4) **grounded-value (grnd)**: the accuracy of the bounding box for each semantic role, where the predicted bounding box must achieve an IoU value of at least 0.5 with the ground-truth bounding box. 5) **grounded-value-all (grnd-all)**: the accuracy of bounding box across the entire set of semantic roles. Besides, these metrics were presented under three evaluation settings: 1) **Top-1-verb**, 2) **Top-5-Verb** and 3) **Ground-Truth-Verb**, verbs are selected based on the top-1 prediction, top-5 predictions, and corresponding ground-truth, respectively. If the predicted verbs are incorrect in the Top-1/5-verb settings, the other four metrics (value, val-all, grnd, and grnd-all) are considered incorrect.

*4.1.3 Implementation Details.* We used the vision transformer with a base configuration (ViT-B/32) as the default backbone for CLIP. We adopted GPT-3.5-turbo for LLM, recognized as a widely utilized LLM in existing works. More details are left in the **Appendix**.

*4.1.4 Baselines.* We compared our proposed method LEX with four strong baselines: 1) **CLS**, which uses class-based prompts to calculate the text embedding for both verb and noun classification. Leverage verb-centric template as the text input of Grounding DINO for role grounding. 2) **TEM**, which uses the verb-centric template to enhance the original class-based prompts for *verb classification*, compared with CLS. 3) **CLSDE**, which generates general descriptions of each noun category without considering scene and role by LLMs to enhance the original class-based prompts for *noun classification*, different from CLS. 4) **RECODE**, which generates semantic role descriptions as visual cues to enhance the original class-based prompts for *verb classification*, compared to CLS.

### 4.2 Quantitative Comparison Result

We evaluated the performance of our proposed LEX and four strong baselines on the test set of SWiG dataset. From Table 1, we have the following observations: 1) The CLS baseline relying solely on class-based prompts, demonstrates inferior performance, particularly in the prediction of bounding boxes and nouns. 2) In the Top-1 and Top-5 verb settings, TEM exhibits a slight performance advantage over CLS, attributed to the enhanced scene information

Table 1: Evaluation results on the test set of SWiG dataset. Values in gray indicate metrics obtained by the same method as CLS.

| Method | Top-1-Verb | | | | | Top-5-Verb | | | | | Ground-Truth-Verb | | | |
|---|---|---|---|---|---|---|---|---|---|---|---|---|---|---|
| | verb | value | val-all | grnd | grnd-all | verb | value | val-all | grnd | grnd-all | value | val-all | grnd | grnd-all |
| CLS [37] | 30.18 | 4.70 | 0.16 | 3.09 | 0.07 | 55.49 | 9.31 | 0.23 | 3.09 | 0.07 | 13.51 | 0.42 | 9.05 | 0.14 |
| TEM | 30.59 | 4.78 | 0.19 | 3.17 | 0.08 | 56.42 | 9.36 | 0.21 | 5.14 | 0.08 | 13.51 | 0.42 | 9.05 | 0.14 |
| CLSDE [32] | 30.18 | 4.86 | 0.18 | 3.15 | 0.07 | 55.49 | 9.32 | 0.23 | 5.00 | 0.08 | 13.80 | 0.44 | 9.12 | 0.15 |
| RECODE [26] | 30.17 | 4.70 | 0.16 | 3.10 | 0.07 | 55.50 | 9.31 | 0.23 | 5.00 | 0.08 | 13.51 | 0.42 | 9.05 | 0.14 |
| LEX | **32.41** | **9.37** | **1.61** | **7.26** | **0.98** | **58.34** | **17.68** | **3.19** | **13.77** | **2.15** | **29.92** | **4.68** | **23.57** | **3.08** |

Table 2: Effects of each component in verb recognition.

| Verb Explainer | Weighting | Top-1-Verb verb↑ | Top-5-Verb verb↑ |
|---|---|---|---|
| | | 30.18 | 55.49 |
| ✓ | | 32.13 | 57.84 |
| ✓ | ✓ | **32.41** | **58.34** |

Table 3: Effects of grounding explainer.

| Grounding Explainer | Ground-Truth-Verb | | | |
|---|---|---|---|---|
| | value↑ | val-all↑ | grnd↑ | grnd-all↑ |
| | 13.51 | 0.42 | 9.05 | 0.14 |
| ✓ | **13.85** | **0.48** | **9.86** | **0.16** |

Table 4: Effects of each component in noun recognition.

| Filter | Noun Explainer | Refine | Ground-Truth-Verb | | | |
|---|---|---|---|---|---|---|
| | | | value↑ | val-all↑ | grnd↑ | grnd-all↑ |
| | | | 13.51 | 0.42 | 9.05 | 0.14 |
| ✓ | | | 28.51 | 4.06 | 21.71 | 2.58 |
| ✓ | ✓ | | 29.16 | 4.33 | 22.11 | 2.67 |
| ✓ | | ✓ | 29.39 | 4.51 | 23.22 | 3.01 |
| ✓ | ✓ | ✓ | **29.92** | **4.68** | **23.57** | **3.08** |

for verb recognition. 3) CLSDE achieves a slight performance gain in the accuracy of noun classification because general noun descriptions often fail to apply in specific contexts. 4) RECODE fails to improve the accuracy of verb recognition, likely due to the insufficient local information provided by semantic roles (*i.e.*, not clear and detailed enough to understand actions). 5) The proposed LEX exhibits significant performance gains across all metrics compared to all baseline models, *e.g.*, **32.41** and **58.34** in terms of verb accuracy under Top-1-Verb and Top-5-Verb, and **29.92**% noun accuracy under Ground-Truth, respectively. This indicates the effectiveness of using explainers in zero-shot GSR.

## 4.3 Ablation Studies

As aforementioned, each component of our proposed LEX framework can serve as a plug-and-play module for zero-shot GSR. This section[4] ablated all the proposed components on the test split of SWiG dataset [36].

*4.3.1 Key Components in Verb Recognition.* We first investigated the two major elements utilized in *verb recognition*: 1) **Verb Explainer**, which denotes using verb-centric descriptions for verb

[4]More ablation studies are left in the Appendix.

classification (Sec. 3.1.3); and 2) **Weighting**, which denotes the employment of description weighting for the text embeddings of all generated verb-centric descriptions (Sec. 3.1.2). Results are shown in Table 2. The first row corresponds to the performance of the CLS baseline. As seen, CLS achieves 30.18% top-1 and 55.49% top-5 verb accuracy. Upon applying the proposed verb explainer (the second row), we observe consistent and substantial improvements for both top-1 accuracy (30.18% → **32.13**%) and top-5 accuracy (55.49% → **57.84**%). This highlights the importance of our "recognition with explainer" strategy and validates the viability of our motivation. Moreover, LEX achieves better performance across all metrics with the weighting strategy. This indicates that the proposed verb explainer and weighting strategy can work synergistically.

*4.3.2 Key Component in Semantic Role Grounding.* We next studied the impact of our **Grounding Explainer**, which adopts rephrased verb-centric templates as grounding input (Sec. 3.2.1). As illustrated in Table 3, our grounding explainer proves to be effective, *e.g.*, **0.81**% accuracy improvement in terms of the grounding (grnd) metrics.

*4.3.3 Key Components in Noun Recognition.* We further analyzed the influence of three major components proposed for *noun recognition*: 1) **Filter**, which denotes filtering those unreasonable nouns by using LLMs (Sec. 3.3.1); 2) **Noun Explainer**, which denotes using scene-specific noun descriptions for noun prediction (Sec. 3.3.3); and 3) **Refine**, which denotes using contextual information of the scene to refine final results (Sec. 3.3.4). From the results in Table 4, we have the following findings: 1) Filtering those unreasonable noun candidates leads to significant performance gains across all noun metrics(*e.g.*, **15**% in value metrics and **3.64**% in val-all metrics). 2) Substantial improvements can be made by incorporating the noun explainer which aims to provide more contextual information (*e.g.*, 28.51% → **29.16**% in value metrics). 3) Furthermore, after incorporating the refine module, we achieve considerable gains of **0.88**% in value metrics and **1.51**% in grnd metrics. 4) Finally, by integrating three core components, LEX delivers the best performance across all metrics. This validates the effectiveness of our comprehensive modular designs.

*4.3.4 Different Architectures of CLIP.* Last, we examined the impact of the utilized CLIP's architectures. As outlined in Table 5, regardless of the visual encoder employed, our LEX demonstrates consistent and substantial improvements across all metrics. Such impressive results further verify the robustness of our approach.

## 4.4 Qualitative Comparison Result

We visualized CLIP's attention maps for various images and query prompts in Figure 6. As is displayed in Figure 6 (a), we can observe

**Table 5: Ablation studies on different architectures of CLIP.**

| Architecture | Method | Top-1-Verb | | | | | Top-5-Verb | | | | | Ground-Truth-Verb | | | |
|---|---|---|---|---|---|---|---|---|---|---|---|---|---|---|---|
| | | verb | value | val-all | grnd | grnd-all | verb | value | val-all | grnd | grnd-all | value | val-all | grnd | grnd-all |
| ViT-L/14 | CLS | 38.69 | 6.50 | 0.16 | 4.23 | 0.02 | 64.65 | 10.95 | 0.22 | 4.23 | 0.02 | 15.65 | 0.38 | 10.24 | 0.10 |
| | LEX | **42.00** | **13.06** | **2.60** | **10.26** | **1.76** | **68.35** | **21.40** | **4.25** | **16.40** | **2.83** | **30.37** | **5.39** | **23.62** | **3.54** |
| ViT-L/14@336px | CLS | 40.04 | 6.69 | 0.17 | 4.41 | 0.05 | 65.67 | 11.03 | 0.19 | 4.41 | 0.05 | 15.38 | 0.31 | 10.49 | 0.11 |
| | LEX | **42.63** | **13.66** | **2.94** | **10.67** | **1.98** | **69.18** | **22.32** | **4.57** | **17.18** | **3.04** | **31.54** | **5.83** | **24.02** | **3.85** |
| ViT-B/32 | CLS | 30.18 | 4.70 | 0.16 | 3.09 | 0.07 | 55.49 | 9.31 | 0.23 | 3.09 | 0.07 | 13.51 | 0.42 | 9.05 | 0.14 |
| | LEX | **32.41** | **9.37** | **1.61** | **7.26** | **0.98** | **58.34** | **17.68** | **3.19** | **13.77** | **2.15** | **29.92** | **4.68** | **23.57** | **3.08** |
| ViT-B/16 | CLS | 32.68 | 5.08 | 0.19 | 3.36 | 0.05 | 58.41 | 9.80 | 0.22 | 3.36 | 0.05 | 14.06 | 0.58 | 9.45 | 0.16 |
| | LEX | **35.11** | **10.03** | **1.93** | **7.74** | **1.19** | **64.18** | **18.29** | **3.71** | **14.05** | **2.44** | **28.96** | **5.01** | **22.72** | **3.16** |

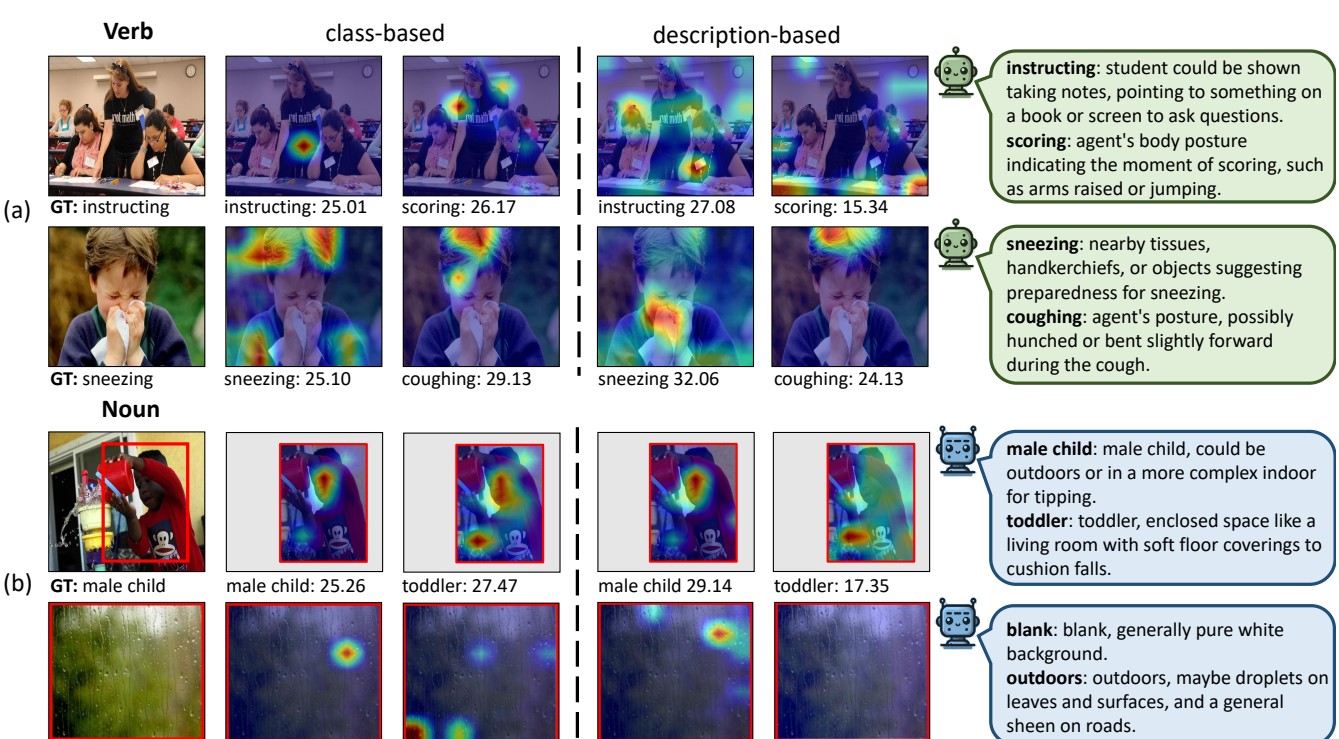

**Figure 6: Visualization of CLIP's attention maps on input images with different prompts. (a) Examples of verb recognition. The right side shows the general verb-centric description prompts generated for each verb, which are used to visualize the corresponding attention map. (b) Examples of noun recognition. The red box indicates the bounding box corresponding to the semantic role. The right side shows the scene-specific noun description prompts generated for each noun.**

that class-based approaches might focus on areas irrelevant to the query verb. For example, given the class-based prompt for the verb "*sneezing*", CLIP wrongly attends to the child's hair and arms. With the detailed guidance of verb-centric descriptions, CLIP can focus on the correct areas, which indicates the importance of providing more contextual information instead of only class names. Similar conclusions can be drawn in terms of noun recognition, as depicted in Figure 6 (b). More qualitative results are left in the **Appendix**.

## 5 CONCLUSION

In this paper, we proposed a novel approach LEX for zero-shot GSR utilizing large language models to provide rich, contextual

explanations at each critical stage of the GSR process. By innovating beyond the constraints of conventional class-based prompt inputs, our approach leverages LLMs to clarify ambiguous actions, accurately ground semantic roles, and ensure context-aware noun identification. Furthermore, our framework enhances model interpretability and adaptability across varied visual scenes without requiring direct training on annotated datasets. Additionally, we introduced a description weighting mechanism to measure the contribution of generated descriptions and assign corresponding weights, thereby enhancing the discriminative power of the method. Extensive validation on the SWiG dataset affirmed the effectiveness and interoperability of our proposed LEX.

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
