# OpenReview forum: "Seeing Beyond Classes: Zero-Shot Grounded Situation Recognition via Language Explainer"
_acmmm.org/ACMMM/2024/Conference — MM2024 Poster_

### Official Review · Reviewer_pYAt · 2024-05-24

**Rating:** 5
**Confidence:** 3

**Summary:**

The authors proposed a novel approach LEX for zero-shot GSR utilizing large language models to provide rich, contextual explanations at each critical stage of the GSR process. By innovating beyond the constraints of conventional class-based prompt inputs, our approach leverages LLMs to clarify ambiguous actions, accurately ground semantic roles, and ensure context-aware noun identification.

**Strengths:**

The motivation of this paper is very clear. The framework of this paper is very intuitive.

**Limitations:**

(1) In Table 1, TEM lacks literature citations.
(2) It is suggested to add some more methods for comparison to demonstrate the cutting edge of the proposed method.

**Suitability:**

2

---

### Official Review · Reviewer_NT9n · 2024-05-24

**Rating:** 4
**Confidence:** 2

**Summary:**

The paper proposes a training-free zero-shot GSR framework LEX to introduce LLM, enhancing the understanding of visual scenes. LEX introduces three explainers. Extensive experiments on the SWiG dataset demonstrate the effectiveness and interpretability of LEX.

**Strengths:**

1. Easy understanding.
2. The paper proposes three explainers to prompt zero-shot GSR.
3. The good experimental results demonstrate the performance surpasses that of previous state-of-the-art methods.

**Limitations:**

1. Introducing the LLM mat increases inference time. Please provide more effectiveness analysis, e.g., inference time.
2. The text generated by large language models may introduce additional noise. The author how to ensure that it is consistently better than the original text.
3. The analysis of some hyperparameters is missing.

**Suitability:**

2

---

### Official Review · Reviewer_tvvz · 2024-05-24

**Rating:** 3
**Confidence:** 4

**Summary:**

The paper propose a new method for zero-shot grounded situation recognition in images. They introduce a novel training-free zero-shot GSR framework LEX that incorporates LLMs as explainers, enhancing understanding of complex visual scenes. The proposed method not only understands the coarse level classes, but also the holistic scene through verbs, semantic roles, their nouns, and their grounding in a zero shot fashion. They evaluate the method on the SWIG dataset.

**Strengths:**

A novel zero-shot method for a complex task which generally requires complex and structured supervision in terms of - verbs, semantic roles, nouns, and groundings.

Solving a complex structured task with LLMs as planners without requiring expensive annotations is inspiring. This presents a new direction for future research in holistic scene level understanding with limited supervision.

**Limitations:**

Performance of the proposed zero-shot approach is quite low when compared with the supervised counterpart. The gap seems too big to be closed. Specially the value and value all metric are way below the SOTA.

The performance on value and value-all is very low, what is the random performance?

The method seems to work well for verbs but not for semantic role labels and noun. Which is the main proposal on the paper- going beyond simple scene level classes. All the complexity of the approach for noun prediction and grounding doesn't seem worth it.

What is the reason behind such low performance? With the current performance, zero-shot SRL doesn't seem like a convincing direction.

**Suitability:**

3

---

### Meta-Review · Area_Chair_TpJL · 2024-06-27

**Recommendation:** Accept (Poster)
**Confidence:** 5

**Metareview:**

All reviewers think the motivation and idea is novel and inspiring. Particularly, reviewer tvvz think the performance gap between zero-shot GSR and the supervised counterpart is large, thus does not think the zero-shot direction is not convincing. After checking the comments and author’s rebuttal, the AC think there are some research and practical values for the proposed zero-shot GSR task, thus encouraging the future research on this direction. Acceptance is recommended.